# Regulation of Notch1 Signalling by Long Non-Coding RNAs in Cancers and Other Health Disorders

**DOI:** 10.3390/ijms241612579

**Published:** 2023-08-08

**Authors:** Joanna Kałafut, Arkadiusz Czerwonka, Karolina Czapla, Alicja Przybyszewska-Podstawka, Justyna Magdalena Hermanowicz, Adolfo Rivero-Müller, Lidia Borkiewicz

**Affiliations:** 1Department of Biochemistry and Molecular Biology, Medical University of Lublin, Aleje Raławickie 1, 20-059 Lublin, Poland; joannakalafut@umlub.pl (J.K.); arkadiuszczerwonka@umlub.pl (A.C.); karolinadudziak@umlub.pl (K.C.); alicja.przybyszewska-podstawka@umlub.pl (A.P.-P.); 2Department of Pharmacodynamics, Medical University of Bialystok, Mickiewicza 2C, 15-222 Bialystok, Poland; justyna.hermanowicz@umb.edu.pl; 3Department of Clinical Pharmacy, Medical University of Bialystok, Waszyngtona 15, 15-274 Bialystok, Poland

**Keywords:** lncRNA, microRNA, Notch signalling, NOTCH, cancer

## Abstract

Notch1 signalling plays a multifaceted role in tissue development and homeostasis. Currently, due to the pivotal role of Notch1 signalling, the relationship between NOTCH1 expression and the development of health disorders is being intensively studied. Nevertheless, Notch1 signalling is not only controlled at the transcriptional level but also by a variety of post-translational events. First is the ligand-dependent mechanical activation of NOTCH receptors and then the intracellular crosstalk with other signalling molecules—among those are long non-coding RNAs (lncRNAs). In this review, we provide a detailed overview of the specific role of lncRNAs in the modulation of Notch1 signalling, from expression to activity, and their connection with the development of health disorders, especially cancers.

## 1. Introduction

The Notch receptors belong to a highly conserved family of transmembrane receptors responsible for transmitting intracellular signals upon cell-to-cell (juxtacrine) contact. One of the most characteristic features of the Notch receptors is that they act as membrane-anchored mechanosensing receptors coupled with a specific nuclear transcriptional modulator—the Notch intracellular domain (NICD) [1]. Notch ligands (Delta-like type ( DLL) and Jagged/Serrate proteins (JAG)), presented by a signal-sending cell, activate the Notch receptors (NOTCH-1, -2, -3, and -4) on the signal-receiving cell [2]. The canonical Notch signalling pathway involves two neighbouring cells; the receptor–ligand interaction results in mechanical forces that unveil a receptor’s proteolytic site near the plasma membrane, recognised and cleaved by the ADAM family of metalloproteinases (Site-2 cleavage; S2), triggering a series of proteolytic events that lead to the release of the NICD, upon gamma-secretase (γ-secretase) cleavage (Site-3; S3) [3]. The NICD is subsequently translocated to the nucleus, where it binds CSL (also known as CBF-1/RBP-jκ, Su (H), or Lag-1) and forms a transcriptional complex that regulates the expression of downstream genes, such as *HES1* (*Hes family bHLH transcription factor 1*), *HEY1* (*Hes-related family bHLH transcription factor with YRPW motif 1*), *MYC* (MYC proto-oncogene, bHLH transcription factor)*,* and *NRARP* (*NOTCH-regulated ankyrin repeat protein*) [4]. The final gene expression patterns and cell fate after Notch activation are highly varied and dependent on various factors such as the cell type, environment, pattern of activation, concentration of receptors and ligands on cell surfaces, and even the duration and strength of activation [5,6]. Such a plasticity in Notch signalling transmission is largely the result of modulation at multiple levels.

The earliest discovery of NOTCH physiological function was the characteristic “notches” found in the distal tip of *Drosophila melanogaster* wings in *Notch*-null mutants, among others, by Thomas Hunt Morgan, Calvin B. Bridges, and John S. Dexter at the beginning of the 20th century [7,8]. Since then, and thanks to advances in genetics and molecular biology, the Notch signalling system has been rapidly unravelled, mainly in *D. melanogaster* and *Caenorhabditis elegans* [9]. It is now clear that Notch signalling plays a key role in developmental and physiological functions. Some of the best described roles of NOTCH1 are the maintenance of stem cell populations [10] and the hippocampal [11] and olfactory [12] plasticity of the nervous systems. Yet it also plays defined functions in angiogenesis [13,14,15,16,17,18] and osteogenesis [19,20], among multiple other developmental processes. The Notch system is important in the regulation of embryonic development, such as during lateral inhibition and induction, binary cell fate, and boundary formation [21,22]. In fact, the Notch pathway is already active in mice embryos that consist only of four cells [23], and it is present in a variety of tissues and organs, such as the thymus [24], vascular system [25], and bone tissue [26], throughout the entire life span. The regulation of Notch activity is performed by transcriptional enhancers, silencers, and enzymes introducing posttranslational modifications, as well as partnering molecules. Aberrations of the multifaceted Notch system result in multiple health disorders, including cancers.

In humans, activating mutations of NOTCH1 was first associated with the development and progression of leukaemia (see also 2.1) [27,28,29,30]. High NOTCH1 activity has been also linked with several other types of cancers including lymphomas [27,28,29,30] and brain [31], breast [32], lung [33,34], ovarian [35], renal [36], and hepatocellular cancer [37,38]. The relationship between the Notch signalling pathway and cancer progression is complex and not only related to an acceleration of cell proliferation. NOTCH1 target genes are also associated with the control of apoptosis, epithelial–mesenchymal transition (EMT), increased drug resistance [39,40,41,42,43,44], modulation of the tumour microenvironment, and maintenance of the cancer stem cell population [45,46]. For these reasons, excessive activation or inhibition of Notch signalling can lead to a serious dysregulation of homeostasis. NOTCH1 activity has both oncogenic as well as tumour-suppressive effects on cells during oncogenesis, depending on the tumour type (for review, see [47]). The oncogenic role of NOTCH1 signalling is also connected with the dysregulation of cellular metabolism and genome stability [48,49].

The dynamic fine-tuning of NOTCH signalling is achieved by post-transcriptional and post-translational regulation. The first includes the control of RNA stability by RNA-binding proteins and microRNAs [50,51]. For example, *NOTCH1* mRNA stability is modulated by the methylation of adenine (m^6^A) residues by N^6^-methyladenosine methyltransferases such as METTL3 and METTL14 [52,53]. These modifications are recognized by proteins such as YTHDF2 (YTH *domain family 2*) that cause the inhibition of Notch signalling by the downregulation of *NOTCH1*, *HES1*, and *HES5* mRNA levels [50].

On the other hand, NOTCH1 is regulated via post-translational modifications (PTMs), meaning an addition of different functional groups, to the target amino acid side chain, which reversibly modulate the structure, activity, localisation, and stability of the target. NOTCH1 is regulated by several PTMs including proteolysis, phosphorylation, acetylation, methylation, hydroxylation, sumoylation, ubiquitination, and *O*-glycosylation [54]. For example, tumour-suppressor cyclin C-dependent kinases, CDK3, 8, and 19, phosphorylate the N1ICD at multiple sites, triggering the binding of Fbw7 ubiquitin ligase and further polyubiquitination and proteolytic degradation [55]. Many kinases also directly control N1ICD transcriptional activity, one of which is casein kinase 2 (CK2). First, it targets S1900 and, further, T1897 of the N1ICD, leading to a decrease in binding between the N1ICD and MAML (Mastermind)–CSL complex, therefore lowering transcriptional activity [56]. Additionally, the phosphorylation of the N1ICD at the second NLS is important for both nuclear localisation [57,58] and transcriptional activity [57], while murine double minute 2 (MDM2)-mediated ubiquitination was shown to enhance N1ICD transcriptional activity [59]. 

The regulatory mechanisms of Notch signalling also include the activity of ncRNAs, which encompass constitutively expressed ribosomal (rRNAs), transfer (tRNAs), small nuclear (snRNAs), and small nucleolar (snoRNA) RNAs, telomerase RNA (TERC), tRNA-derived fragments (tRFs), tRNA halves (tiRNAs), and also regulatory ncRNAs including microRNAs (miRNAs), Piwi-interacting RNAs (piRNAs), small interfering RNAs (siRNAs), circular RNAs (circRNAs), enhancer RNAs (eRNAs), and long non-coding RNAs (lncRNAs) [60]. The interaction between miRNAs and the Notch pathway has been widely reported and reviewed (for example, [61,62,63]); therefore, here, we will focus on lncRNAs and summarise how they affect transcriptional, translational, and post-translational Notch1 signalling.

## 2. Non-Coding RNAs as a Pivotal Control Factor of Notch Signalling

Numerous studies have documented the significant contribution of ncRNAs in the regulation of NOTCH1-related genes, including tumour suppressors and oncogenes. NcRNAs are roughly classified based on the length of the nucleotide sequences, ncRNA position relative to the target gene, and/or ncRNA function. The small ncRNAs (sncRNAs), of less than 200 nucleotides in length, include mainly miRNAs, siRNAs, piRNAs, tRFs, and tiRNAs, while lncRNAs are transcripts with lengths exceeding 200 nucleotides that are usually synthesised by the antisense transcription of target genes (antisense lncRNA) or within intergenic (lincRNA) or intronic loci [64,65]. LncRNAs assist in the remodelling of the genome architecture, gene transcription, and post-transcriptional RNA processing via a direct interaction with other nucleic acids and proteins [66]. As a result, they are involved in processes such as the control of the transcription [67] and translation [68], as well as the formation of nuclear subcomponents by DNA looping [69] and chromatin organisation [70]. Mechanistically, lncRNAs act as scaffolds or decoys for other RNAs and proteins to bind to. The ability of lncRNA to specifically interact with other nucleic acids is harnessed for the formation of transcriptional and chromatin modification complexes, and they also act as sponges for other ncRNAs. Subsequently, lncRNAs supervise other ncRNAs by controlling miRNA availability, mRNA maturation, and siRNA formation. All of these activities allow lncRNAs to oversee gene expression by controlling the function, mutual interactions, and intracellular localisation of proteins and other RNAs [71,72,73].

### 2.1. Controlling Transcription of NOTCH1 and NOTCH1-Related Genes in Cancers

The evolutionarily conserved elements of the Notch pathway are presented throughout the entire Animalia kingdom. Phylogenetic studies suggest that the fundamental molecules of the pathway, the NOTCH receptors and ligands, have played essential roles in animal evolution [74]. In humans, four independent *NOTCH* (1, 2, 3, 4) genes located on the 9q34.3, 1p12, 19p13.12, and 6p21.32 chromosomal regions, respectively, encode NOTCH1-4 proteins which are relatively similar in their core structure. The Notch ligands are encoded by five genes: *DLL* (1, 3, 4) and *JAG* (1, 2) [75]. The expression patterns of NOTCH receptor genes and their ligands depend on the specific cellular context and are often altered in a variety of tumours [48,76]. Moreover, in feedback regulation, activated NOTCH1 affects its own expression as well as expression of other NOTCH receptors and their ligands [77].

Various lncRNAs act as inhibitors or activators of Notch signalling elements. For example, *NOTCH1* expression is regulated, at the transcriptional level, by neighbouring genes, such as lncRNA *RP11-611D20.2* (also known as *LINC01573*) which acts as a *cis* transcriptional activator of NOTCH1 signalling in paediatric T cell acute lymphoblastic leukaemia (T-ALL) and is therefore named *NALT1* (*NOTCH1-associated lncRNA in T-ALL*) (Figure 1(I)) [78]. Recently, a similar role of *NALT1* was reported in gastric cancer (GC), where the knockdown of this lncRNA resulted in a decrease in *NOTCH1* expression, which reduced the invasion and migration of GC cells [79].

The effect of lncRNAs on the chromatin organisation affecting the Notch1 pathway has also begun to be elucidated in breast cancer (MDA-MB-231) cells. The silencing of the highly expressed intergenic lncRNA regulator of reprogramming (*linc-ROR)* led to an increase in NOTCH1, LC3-II (LC3-phosphatidylethanolamine conjugate), Beclin-1, and p53 expression, promoting autophagy and apoptosis [40]. Mechanistically, *linc-ROR* decreases the expression of *miR-34a*, a *NOTCH1* mRNA inhibitor, via the inhibition of histone H3 acetylation in the *miR-34a* promoter region. Similar epigenetic phenomena were described in human cholangiocarcinoma (CCA), in which Enhancer of zeste homolog 2 (EZH2)-mediated histone 3 trimethylation of lysine 27 (H3K27me3) in the same promoter lowered *miR-34a* levels and promoted Notch1 signalling [80].

In the control of gene expression, lncRNAs act also via the regulation of proteins affecting *NOTCH1*. Studies on obesity have shown that NK6 homeobox 1 (Nkx6.1), which binds *NOTCH1* at a 139 bp enhancer sequence (known as the CR2 fragment) in the second intron and positively regulates its expression [81], is upregulated by lncRNA *regulator of insulin transcription ROIT* [82]. *ROIT* interacts with and induces the ubiquitination and degradation of DNA methyltransferase 3a via the proteasome, thereby reducing methylation of the *Nkx6.1* (*NK6 homeobox 1*) promoter and consequently increasing the expression of Nkx6.1 and insulin genes [82]. These data support the importance of the *ROIT*/Nkx6.1/Notch1 pathway in diabetes.

### 2.2. NOTCH1 mRNA and Translation Control

Increasing evidence indicates that both lncRNAs and miRNAs regulate *NOTCH1* mRNA processing. Most of these interactions have been described in stem cells and various diseases including cancer, where aberrant ncRNA levels result in the dysregulation of NOTCH1 signalling.

*NOTCH1* mRNA is tightly controlled by miRNAs binding to its 3′ UTR region; however, lncRNAs can efficiently prevent such interactions as described for *FEZF1-AS1* in non-small-cell lung cancer (NSCLC) [83] and *SNHG7* in breast cancer [84], acting as *miR-34a* sponges sequestering this miRNA from its target mRNAs, consequently increasing the amount of *NOTCH1* mRNA. A similar sponge activity was reported for *NEAT1* (*nuclear-enriched abundant transcript 1*) against *miR-146b-5p* in T-ALL [85], *LncND* (*neurodevelopment*) and *miR-143-3p* in neuronal development [86], *LINC01123* and *miR-449b-5p* in renal cell carcinoma [87], an intergenic lncRNA *346* (*LINC00346*) and *miR-34a-5p* in GC [88], *HCG18* and *miR-34c-5p* in bladder cancer [89], and *DCST1-AS1* binding *miR-92a-3p* in endometrial carcinoma (EC) [90]. Contrarily, lncRNA *CARMEN7* (*cardiac mesoderm enhancer-associated non-coding RNA*), which is increased downstream of NOTCH activation [91], augments *miR-143/145* expression through an enhancer element located in this microRNA’s locus, thereby promoting the differentiation of adult human cardiac precursor cells into smooth muscle cells [92].

A slightly different mode of *NOTCH1* regulation via lncRNAs has been reported in head and neck squamous cell carcinoma (HNSCC) patients, whose low expression of lncRNA *ZFAS1* (*ZNFX1 antisense RNA 1*) correlated with the upregulation of *NOTCH1* and better survival, suggesting an oncogenic role of this lncRNA. Yet the *ZFAS1* levels typically differed depending on the cancer stage and tumour size (T-stage). It was shown that *ZFAS1* overexpression in HNSCC cells and tissues samples is associated with poor patient outcomes [93]. At the molecular level, *ZFAS1* binds *miR-150-5p*, the inhibitor of eIF4E (eukaryotic translation initiation factor 4E), which is required for the translation of several genes involved in proliferation, survival, EMT, and cancer invasion [93]. A year later, studies on the inflammatory response and apoptosis of RAW264.7 macrophages revealed that *miR-150-5p* alleviates those processes, at least partially, via Notch1 targeting. The results highlighted *miR-150-5p* as a target in the development of anti-inflammatory and anti-apoptotic drugs for sepsis treatment [94].

Zhao and collaborators have shown that the presence of *DLX6 antisense RNA 1* (*DLX6-AS1*) is associated with the Notch1 signalling pathway. The knockdown of this lncRNA led to a reduction in Notch1 signalling by diminishing *NOTCH1*, *p21*, and *HES1* at the mRNA and protein levels. Clinical analyses indicated that a high level of *DLX6-AS1* in patients with epithelial ovarian cancer was significantly associated with lymph node metastasis and a poor prognosis [95]. Furthermore, the role of *DLX6-AS1* has been reported as tumour-promoting in pancreatic cancer [96], non-small-cell lung cancer [97], and glioma [98] through inhibiting *miR-181b*, *miR-144*, and *miR-197-5p*, respectively, yet their association with the Notch1 signalling pathway has not been investigated in detail.

In addition to regulation by lncRNAs and miRNAs, the *NOTCH1* transcript can also be backspliced into a circular RNA (circ-*NOTCH1*) during splicing, which has both sponge and decoy activities. For example, circ-*NOTCH1* binds METTL14, depletes the amount of free METTL14, and consequently maintains a high level of *NOTCH1* mRNA [52].

### 2.3. Controlling Notch1 Signalling by lncRNA at the Post-Translational Level in Cancers

LncRNAs often function as scaffolds for signalling proteins, and the Notch pathway is no exception. Several lncRNAs control Notch1 signalling both by directly interacting with the NOTCH1 protein, such as lncRNA *LINC00511* which mediates contacts between the N1ICD transcriptional complex and enhancers, e.g., at the *SOX9* gene [99], or by controlling other proteins that are involved in NOTCH1 activity and processing.

*Neighbour of BRCA1 gene* 2 (*NBR2*) is a lncRNA that directly binds to the N1ICD, as shown in osteosarcoma (OS) [100]. The mechanism of action for *NBR2* is not yet well established; however, it seems to play an essential role in the development of Notch-dependent OS. OS patients with low *NBR2* expression have a shorter overall survival rate compared with patients with higher *NBR2* levels. Additionally, the overexpression of this lncRNA decreased *NOTCH1*, N-cadherin (*CDH2*), and *Vimentin* expression and led to the inhibition of cell migration and proliferation but did not affect apoptosis, both in OS [100] and in NSCLC [101].

As previously mentioned, lncRNAs appear to be strongly associated with the development of T-ALL. The first reports linking human NOTCH1 and carcinogenesis were described in the early 1990s and were related to NOTCH1 chromosomal translocations in T-ALL [102]. Several years later, the genome-wide mapping of NOTCH1-regulated transcripts in T-ALL revealed 182 lncRNAs, among which 55% were interacting with the N1ICD–RBPJΚ (*recombination signal binding protein for immunoglobulin Kappa J region*, also known as CSL) activator complex. One of these lncRNAs is *LUNAR1* (*leukaemia-induced non-coding activator RNA 1*), acting as an enhancer of the expression of its neighbouring gene, *IGF1R* (*insulin-like growth factor1-receptor*), which is essential for T-ALL tumour development in vitro and in vivo [103]. As *LUNAR1* binds to enhancer elements in the promoter of *IGF1R*, its own promoter, and the N1ICD, it was suggested that it exploits the chromatin configuration to recruit the mediator complex and sustain the full activation of the *IGF1R* promoter, fuelling T-ALL development (Figure 1(IIIc)) [103]. In agreement with this, it has been found that the inhibition of *LUNAR1* in CRC decreases tumour growth and efficiently depletes IGF1 pathway activity, indicating a role for the *LUNAR1*/NOTCH1/IGF1R axis in cancer development [104].

LncRNAs may also modulate Notch1 signalling indirectly by binding proteins related to NOTCH1, such as the Paired box gene 8 (PAX8) protein which was proposed to activate the N1ICD in the nucleus by modulating its phosphorylation status and affecting the transcription of its target genes, as well as promoting aerobic glycolysis in pancreatic carcinoma. PAX8 directly interacts with lncRNA *MACC1-AS1* (*MACC1 antisense RNA 1*), increasing its stability, and therefore modulates its activity. The knockdown of *MACC1-AS1* inhibited cancer proliferation and metastasis [105]. Based on these facts, *MACC-AS1*/PAX8/Notch1 signalling might be considered as a target for the alternative treatment of pancreatic carcinoma patients. The recently described functional role of lncRNA interaction with the NOTCH1 receptor is summarised in Figure 2 and Appendix A.

## 3. Notch1-Related lncRNAs and Other Diseases

As the Notch system controls multiple processes, it appears to be tightly related to some congenital and developmental health disorders. Besides cancers, among the dysfunctions linked with abnormal Notch1 signalling are muscle, bone, cerebrovascular, and neurodegenerative diseases. This includes vascular diseases like strokes and ischemic attacks [106], thoracic aortic aneurysms [107], heart malfunction [108], neurodegenerative diseases including Alzheimer’s disease, multiple sclerosis and amyotrophic lateral sclerosis [109], pulmonary dysfunctions [110], severe vertebral abnormalities in spondylocostal dysostosis and spondylothoracic dysostosis [111], disorders related to the dysfunction of muscle, like Duchenne muscular dystrophy [112], bone structure dysfunction, like Klippel–Feil syndrome [113], as well as Hajdu–Cheney [114], Adams–Oliver [115], and Von Hippel–Lindau [116], and multiorgan dysfunction like Alagille syndrome [117]. Still, in many cases, the relation between NOTCH1 signalling and the development or risk of various diseases remains unsettled. Nevertheless, there is mounting evidence linking abnormalities in the lncRNA-mediated control of *NOTCH1* to various levels of its expression in several health disorders (see Table 1).

Among the lncRNAs involved in the epigenetic regulation of Notch1 signalling in diseases are *HOTAIR* and *Potassium voltage-gated channel subfamily Qmember 1 overlapping transcript 1* (*KCNQ1OT1*) [118,119,120]. In autoimmune diseases, the overexpression of *HOTAIR* in both systemic sclerosis (SSc) myofibroblasts and SSc skin biopsies correlates with a reduction in *miRNA-34a* expression and consequent Notch pathway activation. Mechanistically, *HOTAIR* controls the EZH2 methyltransferase-dependent trimethylation of 27 lysine residues in histone H3 (H3K27me3) and therefore epigenetically represses *miRNA-34a* expression [118,121,122]. NOTCH1 is also associated with the development of myocardial infarction (MI). 

**Table 1 ijms-24-12579-t001:** Interaction between lncRNA and Notch1 signalling in health disorders.

Localisation	Disease	lncRNA	NOTCH1 Expression	Cell Line	Animal Models	Patients	Reference
Cardiac system	Acute myocardial infarction	XIST↑	↑	-	AMI rat model	-	[123]
	Calcific aortic valve disease	H19↑	↓	VICs, Saos2, COS7	-	36 patients	[124]
	Myocardial infarction	KCNQ1OT1↑	↑	-	C57BL/6 male mice	-	[120]
	Ischemic stroke	H19↑	↓	-	Male C57BL/6 J mice	40 patients	[125]
Immune system	T-ALL	NALT1↑	↑	Jurkat cells	-	Bone marrow of 20 children	[78]
	Systemic sclerosis (SSc)	HOTAIR↑	↑	-	-	12 adult patients	[118]
Neural system	Epilepsy	NEAT1↑	↑	CTX-TNA	-	6 patients	[126]
	Intervertebral disc degeneration	FAM83H-AS1↑	↑	-	-	10 patients	[127]
	Nasopharyngeal carcinoma	SNHG12↑	↑	SUNE1, CNE1, CNE2 68, HNE-1	-	139 tissue samples	[128]
Head and neck cancer	Oesophageal cancer	MALAT1↑	↑	TE-1, EC109, KYSE30, OE21	-	-	[129]
		SNHG1↑	↑	Eca109, TE-1	-	72 patients	[130]
	Laryngeal cancer	SNHG1↑	↑	-	-	42 patients (different tumour stages)	[131]
Digestive system cancer	Pancreatic carcinoma	MACC1-AS1↑	↑	BxPC-3, PANC-1, MIA PaCa-2, KP-2, AsPC-1, Capan-1	-	2 cohorts (98 and 124 patients) of primary tissues	[105]
	Gastric cancer	NALT1 (LINC01573)↑	↑	SGC-7901, BGC-823	-	336 patients after D2 lymph node dissected gastrectomy	[79]
		LINC00346↑ (sponge for miR-34a-5p)	↑	MGC803, BGC823, MKN28, MKN45, SGC7901	Xenografts in athymic (nu/nu) mice	58 gastric adenocarcinoma tissue samples	[88]
	Colorectal carcinoma	FAM83H-AS1↑	↑	SW480, LoVo, HCT116, HT29	-	40 patients	[132]
		MALAT1↑	↑	COLO205, HCT-116, LoVo, HT26, SW480	Nude Balb/c mice	-	[133]
	Hepatocellular carcinoma	LINC00261↓	↑	SMCC-7721, MHCC97L, MHCC97H	-	66 tissue samples	[134]
Reproductive tract	Ovarian cancer	DLX6-AS1↑	↑	HEY, SKOV3, OVCAR-3	-	128 tissue samples	[95]
		MALAT1↑	↑	A2780, OVCAR3, COC1, A2780/ CDDP, COC1/CDDP, OVCAR3/DDP	-	20 paired tumour tissue samples	[135]
	Endometrial carcinoma (EC)	MEG3↓	↑	HEC-1A, KLE	-	30 tumour tissue samples	[136]
		DCST1-AS1↑	↑	HEC-1	-	62 patients	[90]
	Prostate carcinoma	LINCO1638↑	↑	-	-	42 patients	[137]
		GHET1↑	↑	LNCap, C4-2	-	30 patients	[138]
Other cancers	Breast cancer	linc-ROR↑	↓	MDA-MB-231	-	-	[40]
		SNHG7↑	↑	-	-	37 pairs of tumour tissue samples	[84]
	Lung cancer	NBR2↓	↑	-	-	50 NSCLC patient tissue samples	[101]
		LBX2-AS1↑	↑	A549, PC9, H1975, SPC-A1, H1299	-	165 NSCLC patients	[139]
		EGFR-AS1↓	↑	NCH-H460, NCH-H23	-	87 NSCLC patients	[140]
		LET↓	↑	A549, 95D, NCI-H292, NCI-H1975	-	66 NSCLC patients	[141]
	Osteosarcoma	MEG3↓	↑	MG-63, U2OS	-	-	[142]
		NBR2↓	↑	MG-63, U2OS, SAOS-2	-	62 patients	[143]
		CRNDE↑	↑	MG-63, SAOS-2, U2OS	-	72 patients	[144]

Abbreviations: ↑ represents upregulation; ↓ represents downregulation.

In a mouse MI model, *KCNQ1OT1* was reported to affect Notch1 signalling via decreasing *RUNX3* (inhibitor of Notch1 signalling) expression through the methylation of its promoter [120]. Consequently, the overexpression of *KCNQ1OT1* led to the activation of the Notch1 signalling pathway and the development of MI. In other studies, Zhang et al., reported the regulation of NOTCH expression by X-inactive specific transcript (*XIST*) in acute MI in a rat model. *XIST* silencing led to a lower NOTCH1 expression level via increasing *miR-449*, which manifested in an inhibition of myocardial cell apoptosis and a reduction in the pathological injuries [123]. Thus, targeting *XIST*-NOTCH1 signalling might be considered as a potential therapeutic strategy for MI; however, more detailed research is needed as *XIST* silences the entire X chromosome and the described effect on *NOTCH1* might not be specific.

One of the first discovered eukaryotic lncRNAs, *H19*, was shown to silence *NOTCH1* transcription by preventing the recruitment of p53 to its promoter in calcific aortic valve disease (CAVD). Hadji et al., reported that increased levels of *H19* in CAVD lead to an abnormal mineralisation of the aortic valve [124]. Reducing NOTCH1 signalling in valve interstitial cells increases the expression of genes such as *RUNX2* and *BMP2* which play a pro-osteogenic role [124,145]. A similar mechanism of action of *H19* has been documented in studies of neurogenesis after ischemic stroke. In this case, the silencing of the *H19* lncRNA leads to an increase in *NOTCH1* transcription, which promotes the process of neurogenesis. Moreover, patients with elevated levels of circulating *H19* have a worse prognosis after ischemic stroke [125].

There is also evidence that lncRNAs regulate Notch1 signalling during viral infection. The host NOTCH1 downregulates the prototype foamy virus (PFV) internal promoter activity, which triggers the expression of the viral transcriptional transactivator (Tas), a protein essential for viral replication and gene expression. However, lncRNA *RP5* acts in opposition to NOTCH1 by promoting the expression of *miR-129-5p*, which knocks down *NOTCH1* mRNA, and, therefore, restoring Tas expression. This work provides evidence that some host lncRNAs promote PFV replication by outweighing NOTCH1 inhibition during early viral infection [146].

Data on the regulation of *NOTCH1* mRNA by ncRNAs in health disorders are in abundance. In studies on the development of epilepsy, the silencing of lncRNA *NEAT1* results in the downregulation of *NOTCH1*, *JAG1*, and *HES1*. *NEAT1* affects NOTCH1 signalling by suppressing *miR-129-5p*, which, as described above, targets *NOTCH1* mRNA [126].

## 4. Medicinal Perspectives for lncRNAs in Notch1-Related Diseases

A better understanding of the role of lncRNA on Notch1 signalling might provide new therapeutic strategies. The interconnection between Notch signalling and lncRNAs makes the latter potential biomarkers for Notch signalling activity, as they have high tissue- and tumour-type-specific expression patterns [147,148]. A clear advantage of lncRNAs as biomarkers is their fast and sensitive quantitative analysis by real-time RT-PCR, which can be performed from biological fluids [149,150,151]. The applicability of lncRNAs as biomarkers is well illustrated by *PCA3* (*prostate cancer gene 3*), routinely used in clinical applications as a prostate cancer marker [152], for instance, for patients who obtain a negative result from a prostate biopsy together with a raised level of prostate-specific antigen (PSA) which may indicate undiagnosed cancer [153]. To date, several lncRNAs have been shown to be associated with cancer progression and can be considered as survival factors for certain neoplasias. For example, the upregulation of lncRNAs *DCST1-AS1*, *NALT1*, *LBX2-AS1*, and *MACC1-AS1* correlates with the poor survival of patients with EC, GC, NSCLC, and pancreatic carcinoma, respectively [79,90,105,139]. Meanwhile, the levels of *SNHG1*, *NALT1*, and *CRNDE* are associated with metastases in EC, GC, and OS, respectively [79,130,144]. In colorectal cancer, the expression of lncRNA *CCAT1* (*colon cancer–associated transcript 1*) may potentially predict the response to JQ1, a chemical inhibitor of bromodomain containing four (BRD4) protein important for colorectal cancer proliferation [154]. At the molecular level, BRD4 recognises and interacts with acetylated histone tails, leading to chromatin remodelling by recruiting and stabilising multiprotein complexes to DNA [155,156]. As BRD4 plays an important role in mediating the expression of genes involved in cancers and non-cancer diseases, several drugs targeting it are currently in clinical trials [157].

Among those related to Notch1 signalling, a high *NALT1* expression level has also been detected in T-ALL patient samples [78]. The increased expression of *LINCO1638* lncRNA and *NOTCH1* has been observed in prostate carcinoma patients [137]. Cai et al., showed that NOTCH1-interacting lncRNA *NBR2* expression was downregulated in OS tissues and correlated with a shorter overall survival time compared with patients with higher *NBR2* expression [143]. Due to the specificity of lncRNAs, not only in cancer types but also within subtypes [148], they can serve as biomarkers for patient stratification as well as for drug response prediction. In gastric cancer cell lines (SGC7901/DDP and BGC823/DDP), NOTCH1 was shown to promote the evolution of cisplatin-resistant cells via the upregulation of lncRNA *AK022798* [158], while its downregulation using siRNAs reduced the expression of drug resistance genes. However, subsequent research is required to establish whether lncRNA may be better in predicting the responses of NOTCH inhibitors in relation to what has been discovered so far.

Other important modulators of *NOTCH1* are circular RNAs (circRNAs). Recent studies have shown the potential of circRNAs as prognostic biomarkers, for instance, *hsa_circ_0072309* [159] and *circCNOT2* [160] in breast cancer. Moreover, it has been shown that *NOTCH1* and the Notch1 signalling pathway could be upregulated via *circNFIX*, resulting in glioma progression [161]. *CircNFIX* acts by sponging *miR-34a-5p*, which targets *NOTCH1*. *Circ-ASH2L* also comes in handy in the diagnosis and progression of pancreatic ductal adenocarcinoma, as high *circ-ASH2L* expression was correlated with lymphatic invasion and the TNM (tumour, node, metastasis) stage, plus it was an independent risk factor for pancreatic patient survival. Similar to *circNFIX*, *circ-ASH2L* functions as a miRNA sponge for *miR-34a* and promotes tumour progression in vivo [162].

The circRNA research is still in its infancy, yet with the development of research strategies, effective clinical applications of circRNA will arise and expand in the diagnosis, treatment, and prognosis of cancer.

## 5. Conclusions and Future Perspectives

The efforts to understand the roles of Notch signalling in cancer have revealed various outcomes, either oncogenic or tumour suppressive depending on the cancer type and stage of tumourigenesis. This appears to be largely the result of complex crosstalks between numerous other signalling molecules and pathways. Here, we have reviewed the recent data on the role of lncRNAs in the tweaking of NOTCH1 activity via the regulation of the *NOTCH1* gene, mRNA, and protein. In head and neck tumours, such as HNSCC, where the absence of or reduction in NOTCH1 signalling is beneficial for tumour progression, the low expression of lncRNA *ZFAS1* (necessary for regulation of the translation of several cancer genes) resulted in the upregulation of *NOTCH1* and better survival [93], while in BC, where high NOTCH1 activity fuels tumour progression, *SNHG7* lncRNA promotes the expression of NOTCH1 and EMT by binding *miR-34a*, which causes malignant behaviour and increases the survival and proliferation of BC cells [84]. Therefore, the simple extrapolation of data from studies conducted on other cell types might be misleading in the case of Notch signalling. Also, more research discovering the network between miRNA, lncRNA, and circRNA is needed to further reveal the possible compensatory effects that may affect therapy.

In the case of Notch signalling, it is also important to note that these molecules might act differently on each NOTCH receptor, as each of them gives partially compensatory, partially unique, and partially opposite downstream responses. Thus, lncRNAs seem to be more selective, e.g., for targeting, than other molecular targets such as γ-secretase, which affects all NOTCH receptors and other signalling pathways [163].

Taken together, it is still too early to hypothesise whether lncRNA will be commonly used for therapies or diagnoses. The more we know about the functions of lncRNA, miRNA, and circRNA and their influences on a disease, the closer we are to their routine use in clinics.

## Figures and Tables

**Figure 1 ijms-24-12579-f001:**
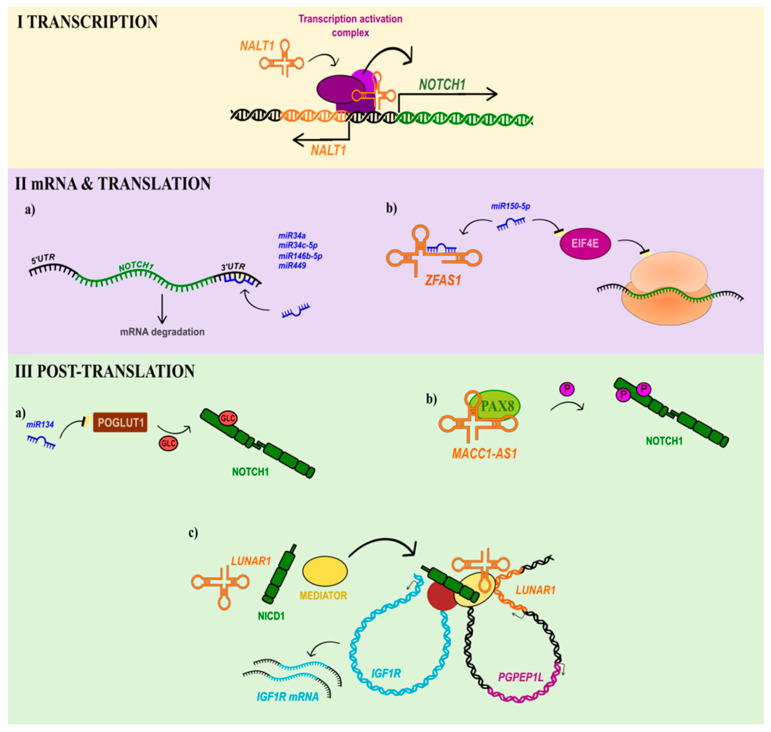
Models of ncRNA action in the control of the Notch1 pathway at the (**I**) transcriptional, (**II**) mRNA and translational, and (**III**) post-translational levels. (**I**) LncRNAs work with various RNA binding proteins (RBPs) to act as a transactivation element for *cis*-regulatory sequences (such as promoters, enhancers, suppressors), e.g., *NALT1* recruits to the *NOTCH1*-promoter transcriptional activation complex promoting Notch signalling in T-ALL. (**IIa**) Several miRNAs have been shown to bind the 3′-UTR of *NOTCH1* mRNA leading to its degradation. (**IIb**) LncRNAs such as *ZFAS1* sequestrate miRNAs, acting as miRNA sponges, i.e., the binding of *miR150-5p* prevents it from inactivating *eIF4E* mRNA and inhibition of the translation of NOTCH1 via blocking the translation initiation step. (**III**) NcRNAs affect the activity and stability of NOTCH1 by regulating the expression of proteins that can modify NOTCH1 protein, such as (**IIIa**) POGLUT1, responsible for NOTCH1 glycosylation, or (**IIIb**) PAX8, controlling the NOTCH1 phosphorylation status. (**IIIc**) LncRNA *LUNAR1* reorganises chromatin in close proximity to its own locus and binds the NOTCH1 and Mediator complex to enhance *IGF1R* transcription.

**Figure 2 ijms-24-12579-f002:**
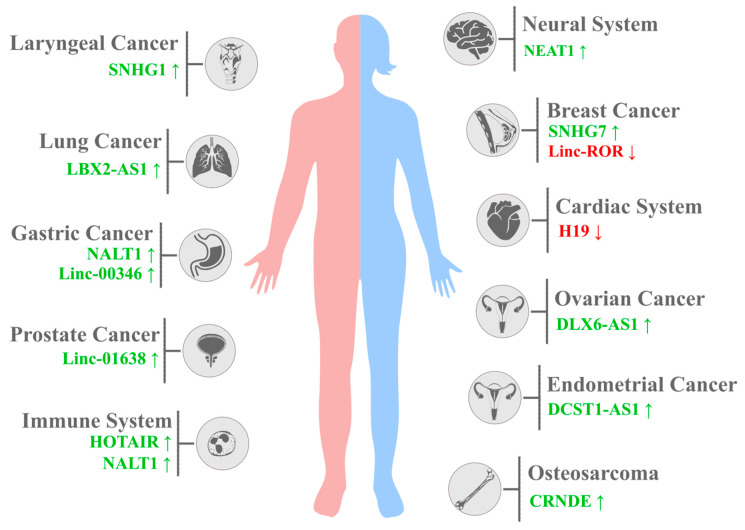
Cancer-related lncRNA overexpression and its impact on the regulation of NOTCH1 expression. LncRNAs upregulating NOTCH1 expression are marked in green and with ↑; lncRNAs downregulating NOTCH1 expression are marked in red and with ↓.

## Data Availability

Not applicable.

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
