# Peer review of "Regulation of Notch1 Signalling by Long Non-Coding RNAs in Cancers and Other Health Disorders"

_ijms, 2023, doi:10.3390/ijms241612579_

Round 1

Reviewer 1 Report

Kalafut et al provides a succinct review of regulation of Notch1 signaling by long non-coding RNAs (lncRNAs) in cancer and other diseases. To this extent, they first provide a brief description of notch1 signaling followed by lncRNAs and their role in the regulation of this specific pathway. The authors also provide the significance of lncRNA-mediated de-regulation of Notch1 signaling in cancer and other diseases.

I believe that the authors provide a well-organized and clear review of lncRNA-mediated regulation of Notch1 signaling in diseases. I believe that the manuscript would be beneficial to the scientists working in the field. However, the following points should be considered prior to warranting publication:

1. As the manuscript includes parts in diseases other than cancer, the title should be revised to represent the content.

2. Lines 101-102, ncRNAs encompass some other RNAs in addition to miRNAs and lncRNAs. Thus it would be better to revise this sentence to imply that other ncRNAs exist as well. The same critique applies to the lines 110-112 as well. Please categorize ncRNAs more accurately. For example, piRNAs, tRFs, rRNA-derived fragments, circRNAs..etc.

3. Lines 312-324, ncRNAs and lncRNAs are used interchangeably, causing confusion to the readers. Please use the correct type of ncRNAs where appropriate.

4. I think that the supplementary table could be a main table.

Minor points:

1. Line 19,  “longnon-coding” should be “ long non-coding”

2. Lines 30-38, please cite a proper reference(s) for these statements

3. Line 110, “ncRNAs” should be “ncRNA”

4. Line 114, delete “the” between “assist” and “remodeling”

5. Line 140, “who” should be “which”

6. Line 161, “began” should be “begun”

7. Line 211, “collaborators” is misspelled

8. Line 219, please remove “yet”

9. Lines 378-379, this sentence is awkward. Please revise.

10. Lines 45 and 54, “cells” should be “cell”

11. Line 164, “what” should be “which”

12. Lines 209-210, anti-apoptosis should be anti-apoptotic

13 Line 216, “have” should be “has”

14. Line 336, “as a survival factors” should be “as survival factors”

Moderate editing of English language required. There are just too many grammar errors. Clarity and cohesion could also be improved.

Author Response

Response to Reviewer 1 Comments

We would like to thank the Reviewer for thorough revision of the manuscript and constructive remarks. Please find below a detailed point by point response to all comments.

Point 1: As the manuscript includes parts in diseases other than cancer, the title should be revised to represent the content.

Response 1: Accoding to the Reviewer’s suggestion we have changed the title for “ Regulation of Notch1 signalling by long non-coding RNAs in cancers and other health disorders”

Point 2: Lines 101-102, ncRNAs encompass some other RNAs in addition to miRNAs and lncRNAs. Thus it would be better to revise this sentence to imply that other ncRNAs exist as well. The same critique applies to the lines 110-112 as well. Please categorize ncRNAs more accurately. For example, piRNAs, tRFs, rRNA-derived fragments, circRNAs..etc.

Response 2:

We agree with this suggestions. We have extended the ncRNAs categorization in Abstract and Introduction.

Point 3:

    Lines 312-324, ncRNAs and lncRNAs are used interchangeably, causing confusion to the readers. Please use the correct type of ncRNAs where appropriate.

Response 3:

We have revised the manuscript according to this remark.

Point 4:

  I think that the supplementary table could be a main table.

Response 4:

We have incorporated the Table into main text.

Point 5:

Minor points

Response 5:

We have corrected manuscript including Reviewer’s remarks.

Point 6:

Moderate editing of English language required. There are just too many grammar errors. Clarity and cohesion could also be improved.

Response 6:

The manuscript was corrected by English language editor.

Reviewer 2 Report

This review by KaÅ‚afut and collaborators provides an overview of the regulation of Notch signalling by long non-coding RNAs. The review is of great interest and is well structured with an optimal subdivision of paragraphs and clear and functional figures. 

I believe that it is appropriate for publication without further modification.

Author Response

We are very grateful for the critical revision of the manuscript.